



# The importance of an estuarine salinity gradient on soil organic carbon stocks of tidal marshes

Marijn Van de Broek[1], Stijn Temmerman[2], Roel Merckx[1], Gerard Govers[1]

[1]Department of Earth and Environmental Sciences, KU Leuven, 3001 Heverlee, Belgium
5   [2]Department of Biology, Ecosystem Management Research Group, University of Antwerp, 2610 Wilrijk, Belgium

*Correspondence to*: Marijn Van de Broek (Marijn.vandebroek@kuleuven.be)





**Abstract**

Tidal marshes are sedimentary environments that are among the most productive ecosystems on earth. As a consequence tidal marshes, and vegetated coastal ecosystems in general, have the potential to reduce atmospheric greenhouse gas concentrations as they efficiently sequester soil organic carbon (SOC). In the past decades, most research has focused on salt

marshes, leaving carbon dynamics in brackish- and freshwater marshes largely understudied and neglecting the diversity among tidal marshes. Moreover, most existing studies underestimate total organic carbon (OC) stocks due to shallow soil sampling, which also influences reported patterns in OC storage along estuaries. We find that SOC stocks vary significantly along the salinity gradient of a temperate estuary (Scheldt estuary, Belgium and The Netherlands), from 46 kg OC m$^{-2}$ in freshwater marshes to 10 kg OC m$^{-2}$ in saltmarshes. In all tidal marsh sediments the OC concentration has a constant value

from a certain depth below the surface downward. However, this concentration decreases with increasing salinity, indicating that the amount of stabile SOC decreases from the upper estuary towards the coast. Although net primary production of macrophytes differs along the estuary, our data suggest that these differences in OC storage are caused mainly by variations in suspended sediment concentration and stable particulate OC (POC) content in the water along the estuary. The fraction of suspended sediments and POC that is transported downstream the maximum turbidity zone is very limited, contributing to

smaller amounts of long term OC sequestration in brackish- and saltmarsh sediments. In addition, high rates of sediment deposition on freshwater tidal marshes in the maximum turbidity zone promote efficient burial of OC in these marsh sediments.

**Keywords**

Tidal marshes, estuarine salinity gradient, soil organic carbon, organic carbon preservation

# 1 Introduction

As a consequence of increasing atmospheric greenhouse gas concentrations and the recognition that soils have the potential to store vast amounts of organic carbon (OC), there is a large interest in the OC storage potential of soils in different ecosystems (Duarte et al., 2013; Govers et al., 2013; Scharlemann et al., 2014). Although coastal vegetated habitats occupy

only 0.2 % of the ocean surface, it has been estimated that they account for approximately 50 % of carbon burial in marine sediments, referred to as blue carbon (Donato et al., 2011; Duarte et al., 2013; Mcleod et al., 2011; Nelleman et al., 2009). It has recently been shown that the OC sequestration potential of these ecosystems will depend to a large extent on future climatic changes and sea level rise (Cherry et al., 2009; Kirwan and Blum, 2011; Kirwan and Mudd, 2012; Weston et al., 2011). Moreover, changing carbon storage in these ecosystems can potentially cause important feedbacks to atmospheric

concentrations of carbon dioxide ($CO_2$) and methane ($CH_4$) (Duarte et al., 2013; Pendleton et al., 2012; Poffenbarger et al., 2011). Constraining the amount of OC that is sequestered in these ecosystems and understanding the processes controlling the size of this pool is of major importance in order to understand the global carbon cycle.





An important fraction of coastal wetlands is occupied by tidal marshes. These are vegetated intertidal areas located along coastlines and estuaries of extratropical regions and are among the most productive ecosystems on Earth (Rocha and Goulden, 2009; Whigham, 2009). Their elevation increases as a consequence of the deposition of both mineral sediments and allochthonous organic matter (OM) during flooding events on the one hand and incorporation of in situ produced belowground biomass on the other hand (Fagherazzi et al., 2012; Neubauer, 2008). Recently formed young tidal marshes, with a low elevation, receive more mineral sediments than their higher counterparts, with sedimentation rates decreasing through time until the marsh platform elevation is in equilibrium with local mean high water level (Temmerman et al., 2003). Despite the fact that the importance of vegetated coastal ecosystems and tidal marshes in particular is now widely recognized, estimates of the total amount of OC stored in tidal marshes are subject to a large uncertainty. Estimates of OC stocks in saltmarshes (i.e. tidal marshes bordering saltwater bodies) range between 0.4 and 6.5 Pg (Bridgham et al., 2006; Chmura et al., 2003; Duarte et al., 2013). To the best of our knowledge, no global estimates are available for brackish and freshwater marshes.

There are multiple reasons for the large uncertainty on estimates of the global OC storage in tidal marsh soils. Firstly, the total area of global saltmarshes currently used to estimate global stocks is poorly constrained, with estimates between 22 000 and 400 000 km² (Chmura et al., 2003; Woodwell et al., 1973), while a global inventory for freshwater marshes is lacking (Barendregt and Swarth, 2013).

Secondly, the dataset available on soil OC (SOC) stocks is limited, both in terms of the number of samples analysed and the geographical scope. Most studies were carried out in a limited number of estuaries, mostly located on the south and east coasts of North America. Differences in sampling procedure and depth beneath the soil surface also contribute to uncertainty. Very often only topsoil samples are analysed, with a limited amount of studies considering carbon storage in deeper horizons, although it has been recognized that these also store a significant amount of OC (Elschot et al., 2015; Wang et al., 2011). An additional factor complicating the extrapolation of data to tidal marshes for which no data is available is the wide range of reported OC contents for tidal marsh soils (Bouillon and Boschker, 2006; Middelburg et al., 1997).

A third important reason for the uncertainties mentioned above is that tidal marshes in estuaries are characterized by steep gradients of multiple environmental and ecological factors (Craft, 2007). First, a sharp increase in salinity towards the coast is present, resulting in a longitudinal estuarine gradient from saltmarshes in the most seaward part over brackish marshes to freshwater tidal marshes. As a consequence of this salinity gradient a vegetation gradient develops, with macrophyte biomass generally being higher on freshwater and brackish marshes compared to saltmarshes (Dausse et al., 2012; Weston et al., 2014; Wieski et al., 2010). Third, the OC input in tidal marsh soils is a mixture of upland, riverine, estuarine and marine sources and the relative contribution of these sources to the total OC pool varies significantly along the estuary (Middelburg and Nieuwenhuize, 1998).

Currently available date suggest that these environmental gradients along estuaries generally result in decreasing SOC stocks with increasing salinity (Craft, 2007; Hatton et al., 1983; Wieski et al., 2010). However, our knowledge on how location along the estuary affects total SOC stocks and which processes control the magnitude of these stocks is, at present, still very





limited. Furthermore, most studies only consider SOC storage down to a depth of 0.3m and generally the reasons for the observed variability are not identified. Nevertheless, understanding the effect of environmental gradients on SOC dynamics in tidal marshes is important. Such understanding may not only help to improve our estimates of current SOC storage but will also be of great help in assessing the effects of sea level rise on these SOC stocks (Morrissey et al., 2014; Poffenbarger

et al., 2011; Weston et al., 2011).

Here, we study the variation in OC inventories in tidal marshes along a salinity gradient in the Scheldt estuary, located in Belgium and the Netherlands. This estuary is characterised by strong gradients in salinity and sediment concentration, making it a suitable location to investigate the impact of these gradients on OC stocks in tidal marsh sediments. The aims of this study are therefore 1) to determine the OC stocks of tidal marsh soils along the salinity gradient of a temperate estuary,

2) to determine the main controls on SOC stocks along this gradient and 3) to use this knowledge to assess how future environmental changes may influence SOC stocks in estuarine tidal marsh soils.

## 2 Materials and methods

### 2.1 Study sites

The Scheldt river is located in Western Europe and flows into the North Sea in the southern Netherlands (Figure 1). The

estuary of the river extends from its mouth up to 160 km upstream where the tide is stopped by sluices near the city of Ghent (Belgium). The estuary is influenced by a semi-diurnal meso- to macrotidal regime, with mean tidal ranges between 3.8 m at the mouth and 5.2 m in the inner estuary (Meire et al., 2005). The estuary has a total length of about 235 km (including tributary tidal rivers) and comprises a salt or polyhaline zone (>18 practical salinity units, PSU), brackish or mesohaline zone (5 – 18 PSU) and freshwater/oligohaline zone (0 – 5 PSU) (Figure 1). The Scheldt estuary is described in detail in Van

Damme et al. (2005) and Meire et al. (2005).

Tidal marshes are present along the entire length of the estuary and tributary tidal rivers, resulting in approximately 498 ha of freshwater marshes, 3035 ha of brackish marshes and 652 ha of saltmarshes (Tolman and Pranger, 2012; Van Braeckel et al. 2013). We sampled a salt, brackish and freshwater tidal marsh, and within each marsh we sampled two locations with different but known rates of historical sediment accretion (Figure 2 and Table 1). The first location was at the high marsh

with an elevation of 0.1 to 0.3 m above mean high water level (MHWL), which has been accreting during the past decades at a rate that is in equilibrium with the rise of MHWL. At the second location, marsh formation started during the past decades at heights well below MHWL. Average accretion rates at these locations were therefore significantly higher than sea level rise (Figure 2). The vegetation history for the different sites is shown in Figure 3. The locations of the sampled tidal marshes are indicated in Figure 1, GPS coordinates of the sampling locations are provided in table S1.





## 2.2 Sample collection

Depth profiles were collected in November 2014 using a manual gouge auger (0.06 m diameter) down to a maximum depth of 1.4 m. Three replicate soil cores were collected with a maximum distance of 3 m in between the coring locations. The cores were divided into 0.03 m sections in the field and transported to the lab. Samples for soil bulk density and root density

measurements were collected using a Kopecky ring sampler or with the gouge auger if soil wetness prevented the use of Kopecky rings. These samples were collected at the soil surface and at depth increments of 0.1 m up to 0.5 m depth, and further down at 0.2 m increments down to maximum depth of 1.4 m. Aboveground biomass was collected on a surface area of 0.25 m² (five replicates) at the end of August at each coring location.

## 2.3 Soil analysis

Before analysis of the soil samples, macroscopic vegetation residues were removed manually using tweezers. The soil samples were oven-dried at 35°C for 48 hours and crushed until they passed through a 2mm sieve. After carbonates were removed with a 10% HCl solution, the samples were analysed for OC, $\delta^{13}C$ and C:N ratio using an Elemental Analyser (FlashEA 1112 HT, Thermo Scientific). Soil texture was determined using a laser diffraction particle size analyser (LSTM 13 320, Beckman Coulter) and grain size was classified into clay (<2 µm), silt (2 – 63 µm) and sand (>63 µm) fractions. Soil

pH was determined after diluting the soil in a 0.01M CaCl₂ solution and electrical conductivity was measured after diluting the samples in de-ionized water.

The collected biomass was dried at 60°C for 48 hours after sediments were removed and weighted in order to calculate the total dry weight of the biomass. The total aboveground biomass of one 0.25 m² surface area was shredded and split until only a small portion was left. This was further grinded and analysed for OC content, $\delta^{13}C$ and C:N ratio using an Elemental

Analyser (FlashEA 1112 HT, Thermo Scientific).

Soil bulk density samples were dried at 105°C for 24 hours. After soil bulk densities were calculated, the samples were washed over a 0.5 mm sieve using de-ionized water and all roots were collected. The roots were cleaned using de-ionized water, dried at 60°C and weighted.

## 2.4 Data analysis

At every location one soil profile was analysed in detail (every other depth sample, i.e. every 0.06 m). For the other two replicate profiles one sample every 0.09 m was analysed down to a depth of 0.72m. Deeper down the profile, one sample every 0.18 m was analysed.

Total SOC stocks were calculated for a volume of soil with a surface area of 1 m² and over the total depth of the sampled marsh sediments. Both the average of the three replicate OC percentages and bulk densities were linearly interpolated to

construct continuous depth profiles.



Root biomass was measured at discrete depths as explained above. For every layer the total root biomass for a surface area of 1 m² was calculated by rescaling the average root biomass for the three replicates to the total volume of that soil layer. Linear interpolation between measurements at different depth intervals was used to calculate the total root density per surface area of 1 m².

To test if aboveground biomass is was significantly different between the sites a one-way analysis of variance was used in Matlab®, after checking for normality using the Anderson-Darling test. For the other variables only three replicates were available so no reliable significance test could be performed.

## 3 Results

### 3.1 Soil characteristics

The studied tidal marsh soils are classified as tidalic Fluvisols with a silt loam texture. The average bulk density ranges from 0.40 to 0.99 g cm$^{-3}$, and both the topsoil pH and electrical conductivity increase from freshwater- to saltmarshes (Table 2).

### 3.2 Vegetation biomass production

Based on the measured maximum annual biomass (figure S1, table S2) and reported values of both above- and belowground annual turnover rates (table S3), annual biomass production for the different sites was calculated, as shown in Figure 4. The average annual aboveground biomass production is the highest for the brackish marshes, followed by the low freshwater marsh and both saltwater marshes. The high freshwater marsh has an aboveground biomass production that deviates from this pattern as a consequence of the fact that only fallen leaves of the willow trees are taken into account at this site, while the woody parts could not be collected, so that we underestimate total biomass production in this case. Upper limits for biomass production on this marsh may be deduced other studies, which typically result in production rates of 500 - 1000 g dry weight m$^{-2}$ y$^{-1}$ (Kopp et al., 2001). No clear pattern in annual production of belowground biomass along the estuary was observed.

### 3.3 Soil organic carbon depth profiles

The depth profiles of SOC show that the depth-averaged concentration decreases from freshwater- to saltmarshes, although the highest topsoil OC concentration is observed at the brackish marshes (Figure 5). In contrast to the freshwater soils, which show a gradual but limited decrease in OC concentration with depth, the brackish- and saltmarshes show a sharp decrease in OC concentration in the top of the profile.



### 3.4 Soil organic carbon inventories

The highest total SOC stocks are found in the freshwater marshes, followed by the brackish- and saltmarshes (Table 3). For every marsh, SOC stocks are greater for the high marshes compared to the low marsh, as a consequence of both deeper marsh soils and higher OC concentrations. In order to compare the marshes directly to each other the stocks down to the largest common depth have been calculated (Table 3). Using this approach, freshwater- and brackish marshes have comparable OC stocks, while both locations on the saltmarsh have significantly lower stocks. Depth profiles of cumulative OC stock per 0.01 m layer are shown in Figure S2.

### 3.5 Stable carbon isotopes

The depth profiles of stable OC isotopes ($\delta^{13}$C) are shown in Figure 6, together with the $\delta^{13}$C signal of above- and belowground vegetation. In general an increase in $\delta^{13}$C values with depth is observed, although deviations from this pattern are observed along the profiles. For all sites except the C4 Spartina site at the low saltwater marsh, the $\delta^{13}$C signal of standing vegetation is closely related to the $\delta^{13}$C signal of SOC in the topsoil layer.

## 4 Discussion

### 4.1 Soil organic carbon stocks along the estuary

The results of this study show that both OC concentrations and stocks of tidal marshes vary significantly along a temperate estuary, with freshwater marshes having the highest stocks, followed by brackish- and saltmarshes (Figure 5 and Table 3). This tendency is in agreement with observations in other studies (Table 4). However, the differences reported in previous studies are almost always much smaller than the differences we find. This may to some extent be related to differences in environmental conditions, but differences in sampling procedures also matter. In most studies, marshes were sampled to a limited depth (Table 4). Generally, the differences in OC content between different marshes are smallest for the top layers. As a consequence, the difference in OC inventory will increase if a larger sampling depth is considered. Evidently, considering a larger sampling depth will also lead to higher estimates of OC stocks. This is one of the factors explaining why our stock estimates are much higher than those reported in the other studies in Table 4, especially for the freshwater marshes. Another issue is whether carbon stocks should be compared by considering stocks down to a certain depth or that the total stock present in the marsh sediments should be taken into account. While it is simpler and more transparent to consider a certain depth, this approach does not account for the differences in dynamics between marshes. As Figure 2 shows, marsh accumulation rates are significantly higher for the freshwater marshes. This automatically implies that, when different marshes are sampled to a common depth, the timeframe that is accounted for will be shorter for those marshes that have the highest accumulation rates (Elschot et al., 2015).





## 4.2 Observed patterns in SOC storage

While our data do not allow for a full statistical or mechanistic analysis of the mechanisms controlling the long-term storage of OC in the studied tidal marshes, some important observation can be made.

A first observation is that low SOC stocks are not systematically related to low biomass production, as no statistical relationship between total annual biomass production (above- and belowground) and SOC stocks is found ($R^2 = 0.01$, figure S3). For example, the annual biomass production at the low saltwater marsh (*Spartina anglica*) is relatively high (Figure 4), while this site is characterised by the lowest SOC stocks. In addition, there is no relationship between annual root carbon production and SOC stocks. This is rather surprising, as it has been proposed that roots contribute significantly to the subsoil OC pool in tidal marshes (Craft, 2007; Saintilan et al., 2013).

A second observation is a very rapid decrease of SOC with depth at the brackish sites. This decrease is accompanied by a shift in $\delta^{13}$C to less negative values with depth in the topsoil of these marshes, suggesting that on the brackish marshes a significant fraction of OC is rapidly decomposed after burial (Figure 6). On the high brackish marsh the decline in OC and the shift in $\delta^{13}$C show the same tendency down to a depth of 0.3 m, while deeper down the profile both variables remain approximately constant with depth. This indicates that a significant fraction (approx. 87 %) of deposited OC is decomposed in this top layer. In the low brackish marsh sediments the situation is different. Here the OC concentration only decreases from the top of the profile down to a depth of 0.15 m, while the $\delta^{13}$C signal increases throughout the profile. At this location *Spartina anglica* was possibly present during early marsh development, resulting in a more positive $\delta^{13}$C signal. This hypothesis is supported by the observation that *Spartina anglica* was indeed present on this marsh before 2000 (Boschker et al., 1999; Middelburg et al., 1997). In contrast, currently *Elymus athericus*, a C3 plant, is present at the marsh surface. This implies that the shift in $\delta^{13}$C with depth at the low brackish marsh could also be the result of a shift from a C4 to C3 type vegetation, rather than resulting from decomposition of OC alone. This is very likely, as in general shifts in $\delta^{13}$C as a consequence of kinetic fractionation during decomposition are in the order of $1 - 3$ ‰ (Choi et al., 2001), while the shift we observe is much larger (5.7 ‰). However, the decrease in OC together with the shift in $\delta^{13}$C in the top 0.15 m suggests that, also on this marsh significant decomposition of deposited OC (approx. 68%) took place after burial.

Also on the high saltmarsh a significant decrease of SOC concentration with depth occurs. This is again accompanied by a shift in $\delta^{13}$C towards more positive values with depth. This location is currently characterised with a mixture of C3 type vegetation. It is uncertain, however, if the isotopic shift with depth can entirely be attributed to kinetic fractionation caused by OC decay. It is likely that at the beginning of marsh growth also *Spartina anglica* was present at this location, as it is currently present at the low part of this marsh. This would imply that also at this location the shift in $\delta^{13}$C with depth is the result of a combination of decomposition of OC and a shift in vegetation from C4 to C3 type.

Our observations indicate that on both the salt and brackish marshes a significant fraction of OC is lost after burial. Although in the brackish marsh sediments a larger fraction of OC is lost after burial compared to saltmarshes, total SOC stocks in the brackish marsh sediments are significantly higher compared to the saltmarshes.





At the freshwater marshes the situation is different. In both the low and high freshwater marsh sediments the decline in OC concentration with depth is very limited. In addition, the $\delta^{13}$C signal does not show a significant shift in the top 0.5 m of the soil profile. Below this depth there is a limited shift in $\delta^{13}$C toward more positive values, but the interpretation of this pattern is complicated by the effect of previous land uses on the marsh (Figure 3). These observations indicate that at both locations at the freshwater marsh there is limited decomposition of OC after burial.

## 4.3 Explanations for the observed patterns in soil organic carbon stocks

An explanation for the variation in SOC stocks between salt and brackish marshes on the one hand and freshwater marshes on the other hand needs to account for the differences in depth gradients in both SOC and $\delta^{13}$C. This may be explained by several factors which are discussed below.

### 4.3.1 Salinity

Although the Scheldt estuary is characterised by a strong salinity gradient (Van Damme et al., 2005), it is unlikely that salinity as such is a direct factor controlling the difference in decomposition of OC, as this would imply that there is a positive relationship between decomposition and salinity. However, litterbag experiments with *Elymus athericus* on a marsh in the Scheldt estuary showed that there was an inverse relationship between soil salinity and decomposition (Hemminga et al. 1991b). In addition, Hemminga et al. (1991b) concluded that there is no significant variation in cellulose decomposition in tidal marsh sediments along the brackish and saltwater portion of the Scheldt estuary.

### 4.3.2 Vegetation type

The type of vegetation present at the different marshes is another possible controlling factor, as it has been shown that different macrophytes have a different resistance against decomposition (Buth and de Wolf, 1985; Hemminga and Buth, 1991; Valery et al., 2004). On the low saltmarsh the presence of *Spartina anglica* is indeed likely to be responsible for the low SOC stocks. While *Spartina anglica* is characterised by a high net primary productivity, the organic material produced is known to be very labile (Boschker et al., 1999; Bouillon and Boschker, 2006; Middelburg et al., 1997).

One of the factors that determine the decomposition rate of plant material is the nitrogen content, as plant material with a higher C:N ratio is generally more resistant against decomposition (Hemminga and Buth, 1991; Jones et al., 2016; Webster and Benfield, 1986). The C:N ratio of the vegetation present at the salt marsh (values between 27 and 30) is significantly lower compared to the vegetation present at the brackish- and freshwater marshes (values between 33 and 55) (Table S2). However, our OC and $\delta^{13}$C profiles suggest that decomposition rates are highest on the brackish marshes and lowest on the freshwater marshes, while the vegetation present at these locations has comparable C:N ratios. Thus, there does not appear to be a direct relationship between the C:N ratio of the biomass and SOC decomposition.





### 4.3.3 Allochthonous organic carbon inputs along the estuary

The OC that is present in tidal marsh sediments is not only derived from autochthonous biomass. Estuaries are often characterised by relatively high concentrations of suspended sediment to which a significant amount of particulate organic carbon (POC) is associated (Abril et al., 2002). Due to the long residence time of water in the Scheldt estuary (2-3 months,

Soetaert and Herman, 1995), organic matter is intensively processed as it moves through the estuary (Abril et al., 2002; Middelburg and Herman, 2007). In addition, mixing between fluvial and marine particles takes place (Nolting et al., 1999; Regnier and Wollast, 1993). Overall, this leads to significant variations in both the quantity and the quality of the POC that is present in the water and that is deposited on the marshes. Clearly, this variation may not only affect the magnitude of the OC inputs but also the decomposability of the OC that is deposited.

The freshwater marshes are located near the upstream border of the Scheldt estuary close to the maximum turbidity zone (MTZ), with average suspended sediment concentrations of ca. 0.15 g l$^{-1}$ (Van Damme et al., 2001; Temmerman et al., 2004). The suspended sediments in this zone contain 7-10% POC (Abril et al., 2002). The higher values are observed in summer, when phytoplankton growth is important, while the lower values are reported in winter. The POC that is present in winter may be assumed to be processed POC from terrestrial origin (Hellings et al., 1999). In addition, during the past

decades a large fraction of OC that has entered the freshwater portion of the estuary originated from untreated wastewater from the city of Brussels (Abril et al., 2002; Billen et al., 2005). It has however been shown that this OC is mineralised on a timescale of weeks, possible even before it enters the estuary (Muylaert et al., 2005; Servais et al., 1987).

Sediment concentrations strongly decline downstream of the MTZ (Abril et al., 2002; Van Damme et al., 2005). At the location of the brackish and saltwater concentrations (ca. 20 km and ca. 50 km from the mouth) sediment concentrations are

about 0.05 g l$^{-1}$ (Van Damme et al., 2001; Temmerman et al., 2004). Furthermore, the POC content of these sediments decreases systematically in the downstream direction, except during the spring season when local production of OC due to phytoplankton is important in the marine portion of the estuary (Muylaert et al., 2005). As a result, average POC concentrations vary between 4 and 6 % in the brackish water zone and between 2 and 5 % in the saltwater zone (Abril et al., 2002). The overall decline in POC content is not only explained by the progressive downstream mineralization of OC but

also by the upstream transport of marine sediments that carry less POC.

The variations in both suspended sediment concentration and POC content have important consequences for the relative importance of allochthonous OC input on the marshes. On the freshwater marshes, both the high suspended sediment concentration and high POC loadings lead to a combination of high sedimentation rates (10-20 mm yr$^{-1}$, with the highest sedimentation rates on the young marshes (Temmerman et al., 2004)) and high inputs of allochthonous POC. On the

saltwater marshes, sedimentation rates are much lower (5-10 mm yr$^{-1}$ (Temmerman et al., 2004)) and the deposited sediments contain 50 – 70 % less OC than the sediments deposited on the freshwater marsh (Abril et al., 2002).

Evidently, these differences may have important effects on OC storage in tidal marsh sediments (Figure 7). It be reasonably assumed that the allochthonous POC that is deposited with the sediments consists for a large fraction of terrestrial,



recalcitrant POC. This POC may be expected to have a high burial efficiency (it will decompose relatively slowly after burial) and will remain in the sediments for a considerable time. The local, autochthonous POC is fresh, will therefore be less recalcitrant and may consequently be expected to decompose much more rapidly with time. Moreover, as both the low and high freshwater marsh are characterised by a very different vegetation (both now and in the past, Figure 3), it is unlikely

that local vegetation contributes significantly to long-term OC storage, given the fact that the SOC concentration is similar at both sites. The decomposition rate of autochthonous POC can be expected to be inversely related to the burial rate, as high sedimentation rates generally promote the burial efficiency of OC (Hartnett et al., 1998; Wang et al., 2014).

Figure 7 illustrates how these factors combine. One may indeed expect to find a much less steep decline of the OC content with depth on the freshwater marsh (Figure 7A) due to (1) the dominance of allochthonous, recalcitrant OC and (2) the rapid

burial of OC. Therefore, a relatively large fraction of labile autochthonous OC is preserved, as it is advected rapidly to deep sediment layers. On the salt and brackish marshes a low sedimentation rate combines with low OC contents of the deposited sediments (Figure 7B). As a consequence, autochthonous OC is a dominant input, but this OC decomposes rapidly with depth. This results in a significant decline of OC content with depth, combined with a significant increase in $\delta^{13}C$ due to kinetic isotopic fractionation.

Thus, both sedimentation rate as well as the rate of allochthonous OC input to the marsh system appear to be important controls on OC preservation in marsh sediments. While other factors such as local biomass production and salinity gradients may also be important, they do not appear to be key controls in the Scheldt estuary as most autochthonous POC appears to decompose rapidly, independent of the specific environmental conditions. This finding is similar to the observations of Omengo et al. (2016), who found that the OC preserved at depth in floodplain sediments of the Tana River in Kenya

consisted dominantly of processed OC that was deposited by the river, while locally produced OC contributed little to long-term OC preservation.

**4.4 Implications of sea level rise for estuarine soil organic carbon stocks**

As global sea level is predicted to continue to rise during the next centuries, progressive intrusion of saltwater further into estuaries may be expected (Robins et al., 2016; Ross et al., 2015). As it is shown that freshwater and brackish tidal marshes

store more SOC compared to saltmarshes (Table 3), one may expect that this will lead to a decrease in OC sequestration at locations where brackish marshes are replaced by saltmarshes. Also the MTZ is predicted to shift more inland (Robins et al., 2016). Because the Scheldt estuary is completely embanked and the tidal wave is stopped by sluices at the city of Ghent, the total area of freshwater marshes is likely to decline after sea level rise (Barendregt and Swarth, 2013). As we have shown that SOC sequestration rates are the largest in the freshwater portion of the estuary, the amount of OC sequestration in the

freshwater portion is likely to decline after sea level rise due to the decline in freshwater marsh area. However, overall sedimentation rates are expected to increase with a rising sealevel, which will automatically lead to an increase in the rate of OC deposition as well as of OC burial rates, resulting in an increase of the OC sequestration rate per unit surface area.

Saltwater intrusion can also influence the decomposition of previously-sequestered OC, with some studies concluding that saltwater intrusion will enhance decomposition of organic matter (Craft, 2007; Morrissey et al., 2014; Weston et al., 2006, 2011), while others find that decomposition rates will decrease (Hemminga et al., 1991a; Weston et al., 2011). From these studies and from the analysis by Chambers *et al.* (2011), it is clear that this effect is highly dependent on local factors, such

as the concentration of elements in the sea water that intrudes the estuary. Therefore, no estimation of the direction of OC mineralisation in tidal marsh sediments following saltwater intrusion in the Scheldt estuary can be made.

The above illustrates that our current understanding of the future evolution of the Scheldt estuary is still insufficient to make a quantitative assessment of how SOC stocks in the tidal marsh environment may change in the future.

**5 Conclusion**

As reported data on estuarine gradients of SOC are very scarce and, more importantly, often based on shallow soil sampling, additional research is needed in order to better constrain estimates of global estuarine OC stocks.

This study shows that the quantification of  SOC stocks in tidal marsh sediments critically depends on the sampling depth. Gradients in SOC concentrations with depth strongly vary between marsh type so that a full inventory can only be made if sampling is carried out over the entire depth of the marsh sediments. Even if such data are available, interpretation has to be

done with care, as sedimentation rates may vary considerably within a single estuary, making it complex to convert inventories to sedimentation or preservation rates.

In the Scheldt estuary, total SOC stocks are largest in a freshwater- and brackish tidal marsh and significantly lower in a saltwater marsh. These variations are to some extent controlled by variations in autochthonous biomass production, but our data strongly suggest that the key control on long-term OC preservation is the relative contribution of allochthonous to total

OC input, while OC burial rate may also be important.

The impact of future sea level rise on OC stocks in tidal marsh sediments will be determined by an interplay of different factors, including the evolution of the spatial extent of marshes in different salinity zones and sediment and OC deposition rates. Our study allowed to identify the factors that are important controls on OC storage and may need further research to resolve this issue.

**Acknowledgements**

We would like to thank Lore Fondu and Jianlin Zhao for their much-appreciated help during field work and lab analysis. We are also grateful to Natuurpunt, Staatsbosbeheer Zeeland and Stichting het Zeeuwse Landschap to provide us the opportunity to sample the tidal marshes.

**Competing interests** The authors declare that they have no conflict of interest.



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



**Table 1: Main properties of the sampled tidal marshes.** [A]from Meire et al. (2005), [B]from Abril et al. (2002), [C]Atriplex portulacoides, Limonium vulgare, Triglochin maritima, Elymus athericus, Puccinellia maritima, [D]based on depth profiles of texture and OC concentration

| Name | Name in this study | Vegetation | Tidal range (m)[A] | Elevation relative to local MHWL (m) | POC% of suspended sediment[B] | Max. marsh sediment depth (m)[D] |
|---|---|---|---|---|---|---|
| Notelaar marsh | Freshwater low | *Phragmites australis* | 5.14 | +0.24 | 6 - 10 | 1.2 |
| | Freshwater high | *Salix sp + Urtica dioica* | 5.14 | +0.25 | 6 - 10 | > 1.4 |
| Waarde marsh | Brackish water low | *Elymus athericus* | 4.85 | +0.01 | 4 - 5 | 0.75 |
| | Brackish water high | *Elymus athericus* | 4.85 | +0.14 | 4 - 5 | > 1.4 |
| Paulina marsh | Saltwater low | *Spartina anglica* | 4.19 | -0.66 | 3 - 4 | 0.2 |
| | Saltwater high | Mixed vegetation[C] | 4.19 | +0.11 | 3 - 4 | 0.6 |



**Table 2: General characteristics of the soil profiles at the studied sites. [A]Average for the upper meter, [B]Value for topsoil only, [C]Up to 0.7m depth, [D]Up to 0.2m depth**

|  | Bulk density (g cm$^{-3}$)[A] | Soil pH[B] | Electrical conductivity (dS cm$^{-1}$)[B] |
|---|---|---|---|
| Freshwater low | $0.40 \pm 0.07$ | $7.47 \pm 0.02$ | $0.0271 \pm 0.0009$ |
| Freshwater high | $0.54 \pm 0.04$ | $7.35 \pm 0.10$ | $0.0262 \pm 0.0007$ |
| Brackish water low | $0.89 \pm 0.06$[C] | $7.70 \pm 0.06$ | $0.0389 \pm 0.0048$ |
| Brackish water high | $0.99 \pm 0.06$ | $7.49 \pm 0.09$ | $0.0365 \pm 0.0023$ |
| Saltwater low | $0.63 \pm 0.07$[D] | $7.93 \pm 0.02$ | $0.0959 \pm 0.0021$ |
| Saltwater high | $0.96 \pm 0.11$ | $7.87 \pm 0.03$ | $0.0113 \pm 0.0010$ |



**Table 3: Total OC stock (kg OC m⁻²) and standard deviations. The depths down to which the stocks are calculated are given between brackets.**

| | OC stock (kg OC m$^{-2}$) | |
| --- | --- | --- |
| | Low marsh | High marsh |
| **For the entire marsh profile** | | |
| Freshwater | 32.35 ± 0.65 (1.2m) | 46.44 ± 0.80 (1.4m) |
| Brackish water | 20.50 ± 0.72 (0.75m) | 32.23 ± 0.31 (1.4m) |
| Saltwater | 2.84 ± 0.10 (0.2m) | 9.93 ± 0.34 (0.6m) |
| **Up to 0.6m depth** | | |
| Freshwater | 16.38 ± 0.54 | 21.66 ± 0.71 |
| Brackish water | 18.63 ± 0.71 | 19.63 ± 0.27 |
| Saltwater | - | 9.93 ± 0.34 |



**Table 4: Reported SOC stocks (kg OC m⁻²) of tidal marsh soils along estuarine salinity gradients. ᴬData for high marshes only**

| Estuary | Sampling depth (m) | Freshwater | Oligohaline | Mesohaline | Polyhaline | Reference |
|---|---|---|---|---|---|---|
| Delaware (U.S.A.) | 0.16 | 3.136 | 2.41 | 3.528 | - | (Weston et al., 2014) |
| Sapelo Doboy, Altamaha (Georgia, U.S.A.) | 0.30 | 8.379 | 10.692 | 4.626 | 5.932 | (Craft, 2007) |
| Dovey (Wales) | 0.10 | - | 2.8 | 1.8 | 2.4 (low), 1.4 (high) | (Dausse et al., 2012) |
| Barataria (Louisiana, U.S.A.) | 0.38 | 10.3 | 24.1 | 12.9 | 12.8 | (Hatton et al., 1983) |
| Satilla Altamaha Ogeechee (Georgia, U.S.A.) | 0.30 | 8.096 ± 1.245 | - | 6.816 ± 0.997 | 6.069 ± 0.482 | (Wieski et al., 2010) |
| Barataria basin (Louisiana, U.S.A.) | 0.50 | 5.37 | - | 4.38 | 2.90 | (Williams and Rosenheim, 2015) |
| San Francisco Bay (California, U.S.A.) | 0.20 | - | - | 7.82 | 5.33 | (Callaway et al., 2012)ᴬ |
| Louisiana (USA) | 1.5 | 65.76 | - | - | 56.65 | (Wang et al., 2011) |



| Scheldt (Belgium, The Netherland) | 0.6 | - | 21.66 ± 0.71 | 19.63 ± 0.27 | 9.93 ± 0.34 | This study[A] |





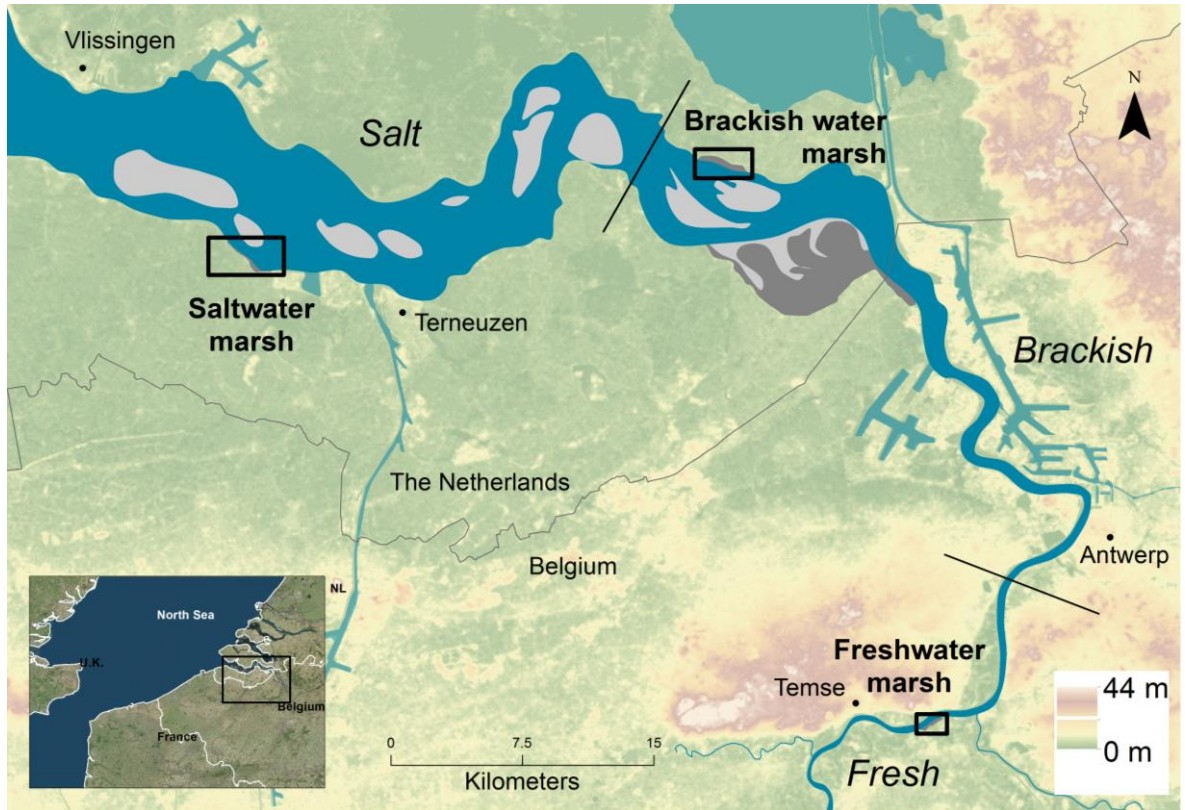

**Figure 1: Map of the Scheldt estuary showing the salinity zones and the location of the sampled tidal marshes in a western European context. Intertidal sandflats are depicted in light grey.**




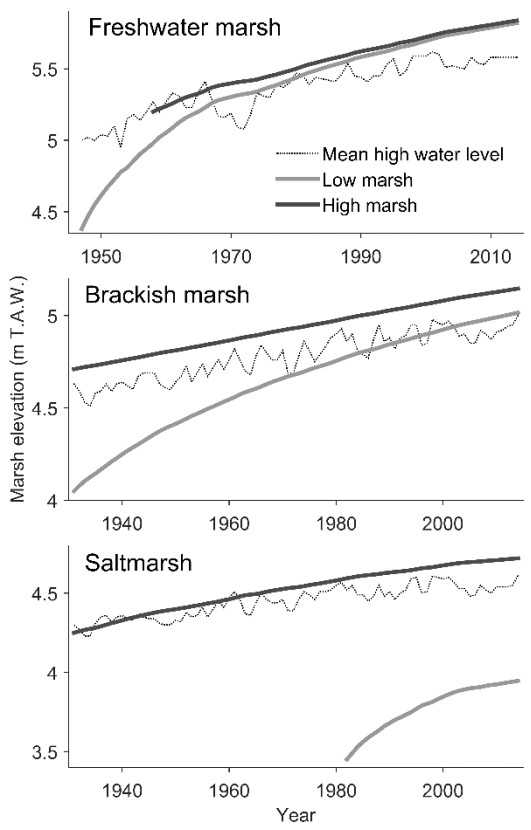

**Figure 2: Evolution of marsh surface elevation and mean high water level (relative to Belgian ordnance level, m T.A.W.) at the sampled locations (based on Temmerman et al., 2004).**





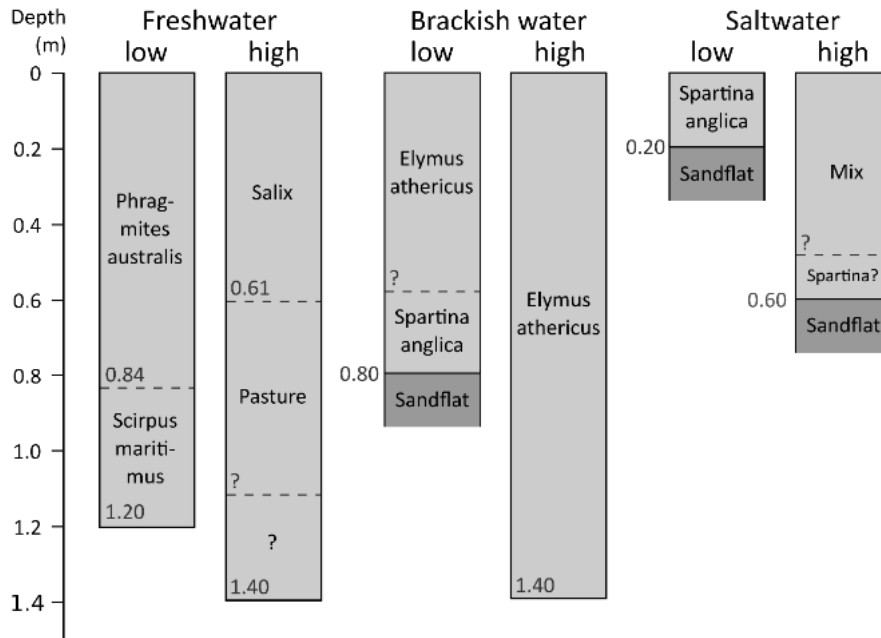

**Figure 3: Depth profiles of the sampled tidal marshes showing the vegetation history at each location. At shallow marshes the former tidal sandflat was reached, at other locations the marsh sediments extended below the maximum sampling depth of 1.4 m. Vegetation history is based on Temmerman et al. (2003) and δ¹³C profiles from this study combined with information from Boschker et al. (1999) and Middelburg et al. (1997).**




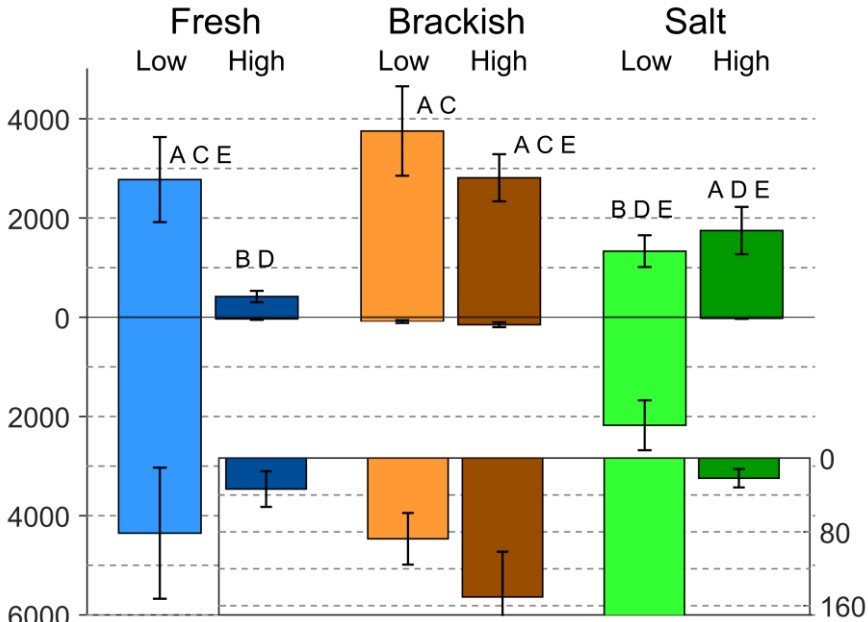

**Figure 4: Annual biomass production (g dry weight m$^{-2}$ yr$^{-1}$) (the inset is a magnification of the root biomass). Standard deviations for aboveground biomass are calculated based on 5 replicas, for root biomass on 3 replicas. Significantly different aboveground biomass is denoted with different letters.**




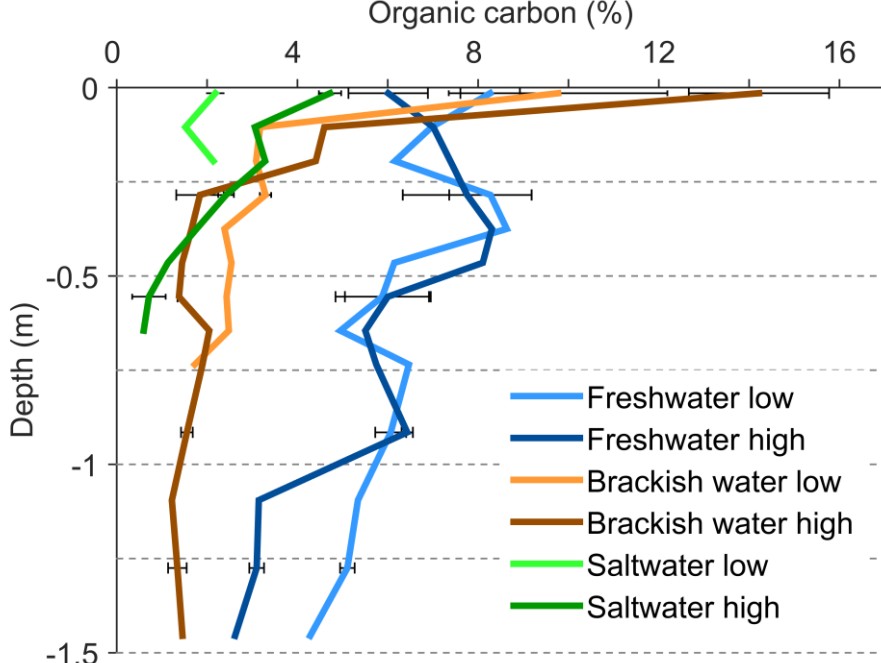

**Figure 5: Depth profiles of OC concentration for all study sites. Data points show the average of three replicate soil samples. Error bars for specific depths are shown and represent the standard deviation of three replicate soil profiles.**





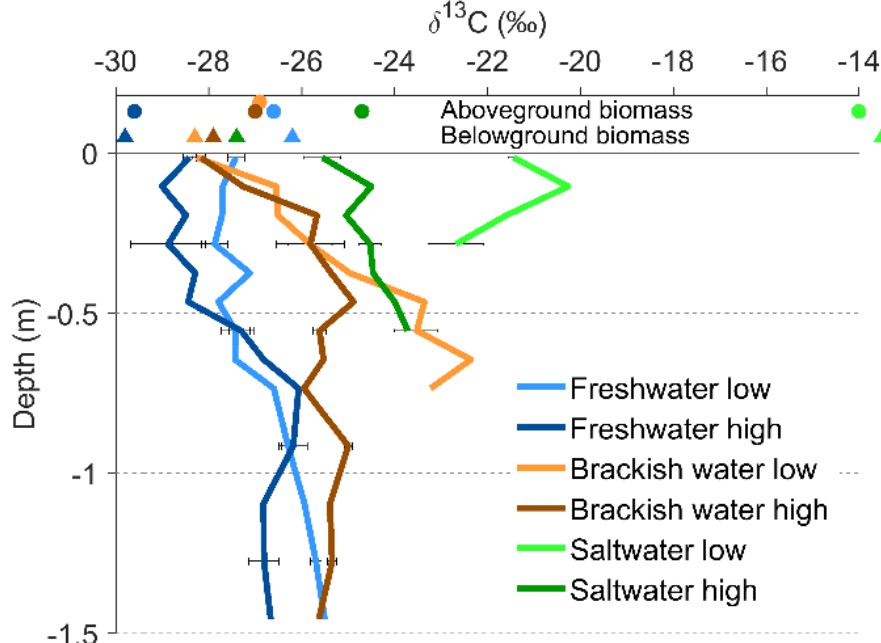

**Figure 6: Depth profiles of δ¹³C, together with the δ¹³C signal of aboveground (cirles) and belowground (triangles) biomass (values are provided in table S1). Error bars represent the standard deviation of three replicate soil profiles.**

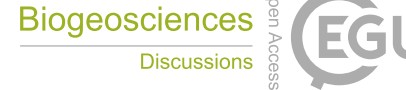

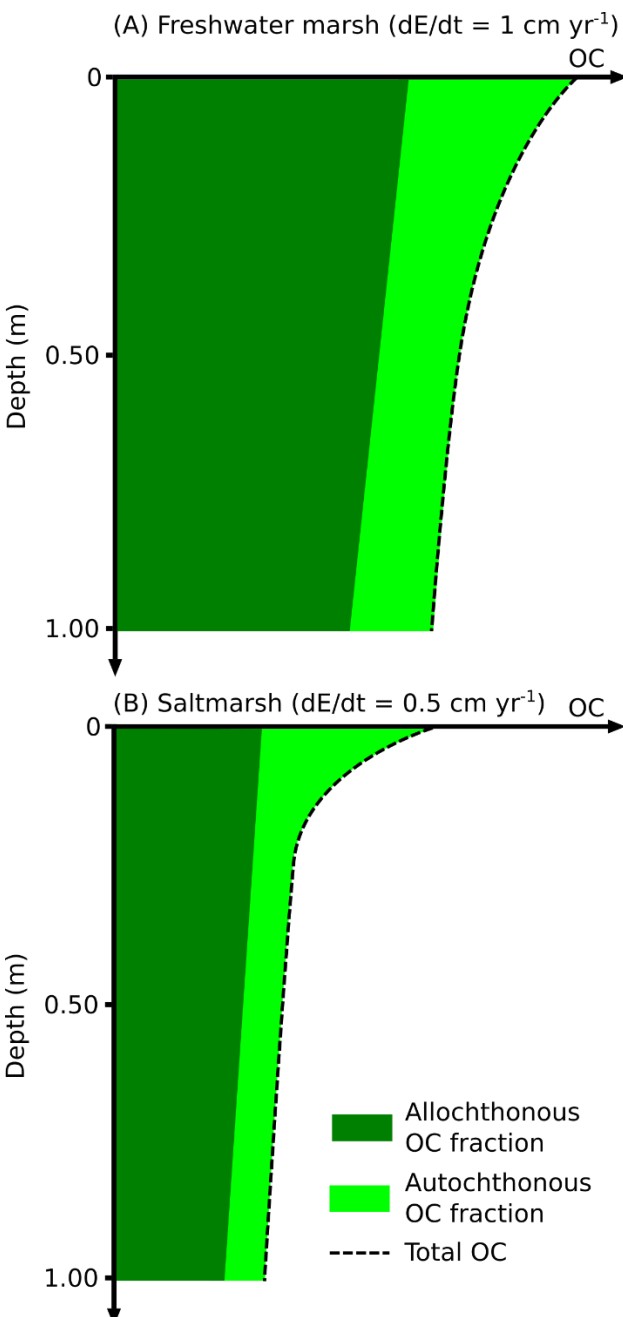

**Figure 7: Conceptual diagram of the effect of both sediment deposition rate (dE/dt, E = elevation) and the relative inputs of recalcitrant allochthonous OC and labile autochthonous OC on the fate of buried OC in a tidal freshwater marsh (A) and saltmarsh (B).**