# Peer review of "The importance of an estuarine salinity gradient on soil organic carbon stocks of tidal marshes"

_Biogeosciences, 2016_

## Referee Comment (RC1) · Anonymous Referee #1 · 30 Aug 2016

General comments: The paper is well-written and generally well structured. It addresses an important gap in the field of carbon cycling, namely of measurements in brackish and freshwater marshes. The authors address various aspects clearly and draw attention to the problems associated with different sampling depths. In addition, they indicate what a future sea level rise may entail for the carbon storage dynamics within the Scheldt estuary.

There are some aspects which need clarification and one main concern of mine is that samples were collected in different seasons. Depth profiles were collected in November whilst aboveground biomass was not collected until August. No mention of this is made in the discussion and I certainly believe that this needs to be addressed and

justified.

I also miss more discussion on the effect the very different types of vegetation may have on the carbon dynamics of the different marshes. This certainly influences stable isotope signatures and carbon mineralization rates. More comments are found in the specific comments regarding this.

Specific comments: P3 L3f: Why did the authors limit themselves to the incorporation of in situ produced belowground biomass? Aboveground biomass also produces substantial amounts of litter and can also be buried.

P4 L18-20: the use of PSU/practical salinity unit is discouraged, nowadays salinity as written here would be unitless i.e. the authors should write: ". . .salt or polyhaline zone (salinity >18), brackish or mesohaline zone (salinity 5–18) and freshwater/oligohaline zone (salinity 0–5). . ."

P5 L4: How were these samples stored during their transport i.e. were the 0.03 sections thus homogenized?

P5 section 2.2: why were depth profiles collected in November 2014 and aboveground biomass not until the end of August? How do the authors justify using data from such different seasons?!

P5 Section 2.3: just make one paragraph for easier reading and change title to "Soil and biomass analysis"

P5 L18: What do the authors mean with split? This also needs clarification because now it sounds like only one of the five replicates was analysed. Is this the case, or are you describing what was done to each one of the five replicates? Please clarify.

P5 L25ff: The authors sectioned the cores into 0.03 m sections, so, when they say one sample every 0.09m, do they mean it is the sample at 0.06-0.09, or 0.09-0.12 and so forth? The same question applies to when they say every 0.18 m. Maybe rather say ". . .For the other two replicate profiles every third sample was analysed (i.e. 0.06-0.09,

0.15-0.18, . . ., 0.69-0.72m) to a depth of 0.72m. Thereafter, a sample was analysed every 0.18 m."

P5 L29: what linear interpolation technique was used to do this?

P6 L6: Was only a check for normality done? Please also mention (and I hope the authors did!) that homogeneity of variance was also checked.

Please also specify what statistical programme was used to run the analysis. What techniques were used for the posthoc analyses as I presume some were done since you mention differences in Figure 4? And please specify which level of probability was used (e.g. "with a level of significance of $p<0.05$).".

P6 section 3.1: - also include here that detailed results for the grain size (not texture) are in the supplementary information

Section 3.2: - Figure S1 is not maximum annual biomass but as is noted in the figure caption as total biomass. This is a difference so please clarify. - Even if the below-ground data was not statistically analysed and no clear patterns are observed, I would have liked to see some comments on what we see i.e. that at the fresh low biomass is clearly very high, that for most sites we see very low values. - An explanation is needed here for figure 4 and the letters apparently showing differences. These need to be explained.

P.7 section 3.4: - Depth profiles of cumulative OC stock per 0.01 m layer are shown. . . Where does this 0.01 m sectioning come from? The authors make no mention of this is in the methods. There you can only find 0.03 m sections or 0.1 and 0.2 m sections. Please clarify what I have missed. - Please be more consistent with the terminology. Within this one paragraph the authors begin by using SOC but then use only OC later on.

P7 L 11f: 13C signal of standing vegetation is closely related to the $\delta$13C signal of SOC in the topsoil layer. How is this conclusion reached? I presume with standing

vegetation you mean the aboveground biomass? I would not agree with this from what I see in figure 6.

P7 L18: "However, the differences reported in previous studies are almost always much smaller than the differences we find. This may to some extent be related to differences in environmental conditions, but differences in sampling procedures also matter." I agree that the authors want to address the problem of inconsistent sampling depth but I do not think that you can dismiss all the other reasons why there are such differences with this one sentence. The estuaries listed in Table 4 are all very different in terms of their geology, morphology, inputs, outputs, etc. and I would like to see some more discussion of this. One of the aims of this paper was to determine OC stocks along a salinity gradient of a temperate estuary and its main controls and I think this has to be addressed more thoroughly. Since the authors do actually discuss some of these factors in section 4.3, I would suggest that section 4.3 follows directly to 4.2 (or is combined) because the authors here try and further explain the observed patterns in SOC stocks which is a more natural progression from what is initiated in section 4.1. I would also bring the issue of different sampling depths then as a separate header and not as the first paragraph of the discussion. This is an aspect but not the most important one.

In relation to this it is unclear in line 20 whether the authors refer to differences from this study or from the other studies. This needs to be clarified.

Section 4.3.2: I miss a more thorough discussion on the fact that you have very different vegetation types. I presume no $\delta$ 13C values are known for the different plants themselves? I also struggle with the fact that biomass was only measured in August, whilst all other measurements were taken in November. The influence of weather and climate conditions and subsequently river flow on affecting stable isotope signatures should not be underestimated (e.g. Zetsche et al. 2011, dx.doi.org/10.1016/j.csr.2011.02.006).

I would suggest the authors also look at a recent similar study by Hansen et al. 2016

(DOI 10.1007/s11368-016-1500-8) and see how their results of the importance of salinity can be reconciled in this study also for section 4.3.1.

P8 L7f: There is no relationship. Did you analyse this statistically? If so please provide test results here, or at least indicate (data not shown).

P8 L19f: Elymus is considered an invasive species. Do you think it is invading here and will remain as the dominant vegetation type here? How will this affect influence SOC stocks in the future as conditions favour this plant?

FIGURES: Personally I would prefer it if the authors used the blue colours always for the saltmarshes (since closest to the blue ocean) and the green colour for the freshwater marshes (closest to land) in the figures. This is more intuitive to the reader.

Figure 1: Please increase the font size of the country names in the inset. FYI: A black and white version of the map will not depict the light grey areas.

Figure 2: Brackish water marsh not just Brackish marsh

Figure 3: All species names should be italicized. Figure caption: At several marshes the former tidal sandflat was reached, whilst at two other locations the marsh sediments extended below the maximum sampling depth of 1.4 m. The vegetation history is based on Temmerman et al. (2003) and information from the $\delta13C$ profiles of this study, in combination with information from Boschker et al. (1999) and Middelburg et al. (1997). Mix denotes a mixed vegetation which included the following species: . . . . A '?' indicates that no clear identification was possible.

It is not possible to say only shallow marshes because the sandflat is also reached at the high saltmarsh and I presume only freshwater and brackish water high went beyond 1.4 m? Also specify what mix stands for. The figure has to be understandable on its own.

Figure 4: the inset is very distracting. Please remove. Instead you can insert a break on the y-scale to allow the details to be seen more easily for the belowground biomass.

Adjust the figure caption i.e. remove "(the inset….biomass)". Also add the y-axis legend i.e. Biomass production (g dry weight m-2 yr-1). Replicas should be replicates. The letters to indicate significant differences are confusing. It has to be explained in the figure caption what the different letters stand for. No mention of these are made in the main text which also has to be addressed!

Figure 5: Error bars for specific depths represent the standard deviation.

Figure 6: aboveground (circles)… Error bars represent the standard deviation.

Figure 7: write out OC once as organic carbon in the figure caption.

TABLES: Comments like A, B, C etc. should be added as footnotes. They are footnotes and should not be in the main caption text.

Table 1: please change around C and D (better to have A, B, C in the same line and then D at the bottom for the mixed vegetation. Please also italicize all species names in the footnote D (previously footnote C). Regarding footnote C (previously D): What is texture? It is not texture but grain size that was measured in this study. Why is this called maximum marsh sediment depth? I would rather simply write "Maximum sampling depth". The tidal sandflat that is reached most likely is deeper but probably caused problems with the sampling device? Sand is not easy to sample.

Table 2: Keep footnotes C and D and make them A and B. Add to figure caption: "Bulk density values are averages for the upper meter of soil, whilst soil pH and electrical conductivity were measured in the topsoil only.

Table 3: Increase the space between the line termed saltwater and the next line for 'up to 0.6 m depth' to make this clearer for the reader. Figure caption: Total organic carbon (OC) stock (kg… deviations calculated for the full vertical sampling profiles (depths used for the calculations are given in brackets), and the upper 0.6 m.

Table 4: make this into a horizontal table and thus more readable. Perhaps place the location then as a separate column next to the estuary name.

Supplemental data: I would welcome that the excel sheets provided in the supplemental data are at least referred to in the paper.

Figure S1: see my comments on Figure 4. Please also remove the inset here.

Figure S2: why is there now mention of a depth interval of 0.01m? This is never mentioned previously in this study, only slicing at 0.03 m and 0.1 +0.2 m intervals is ever mentioned. Please explain.

Table S3: Please italicize all species names. Replace Oosterschelde with Eastern Scheldt and Westerschelde with Western Scheldt.

Table S2: Figure caption: Average values ($\pm$SD) for aboveground, belowground (maximum root depth is given in brackets (m)) and total biomass, biomass production, organic carbon and nitrogen concentration (%), C:N ratio as well as the $\delta$13C signal (‰ for vegetation at the study sites. Remove footnote A, footnote B: write here in full as a footnote the species. In table: Adjust either DW or dry weight, now have both. Also write species names in full. If you miss space you can shorten Freshwater to Fresh, etc. and add to caption "...at the study sites (freshwater, brackish water and saltwater marshes)."

Technical corrections: P2 L14: downstream of the maximum...

P3 L2: replace extratropical with temperate. Extratropical is not normally used in this context.

P3 L7: equilibrium with the local

P3 L8: remove 'in particular'

P3 L16-17: remove spacing and merge into one paragraph.

P3 L22: tidal marshes, for which no data is available, is the

P3 L23-24: remove separation into paragraphs. These three reasons are all one aspect

and should be together in one paragraph.

P3 L25: ... (Craft, 2007). A sharp increase in salinity...

P3 L29: ...2010). In addition, the OC input in tidal marsh...

P3 L32: data not date

P4 L5: remove space and form one paragraph.

P4 L8: ...stocks in tidal marsh soils. The aims...

P5 L6f: ...0.5m depth, and then in 0.2 increments down to the maximum depth of 1.4m.

P5 L17 and L23: replace weighted with weighed. Samples were placed on a scale, hence they were weighed. Weighted is used in a different context.

P5 L19: ...using the Elemental Analyser...

P5 section 2.4: remove line spacing and form one paragraph.

P5 L26: ...analysed to a depth of 0.72m. Below this depth, samples were analysed every 0.18 m.

P6 L5: remove "is"

P6 L17: willow trees were

P6 L18: what is meant by woody parts, this is not a correct term!

P6 L19: deduced from other studies

P6 L23: showed and decreased i.e. past tense.

P6 L26: do not write just in the top of the profile, be more specific, e.g. " ...OC concentration in the upper 0.2 m." Or whichever depth it is...

P7 L2: to the low marshes

P7 L11: this is the first time a 'C4 Spartina site' is mentioned, please refer to this differently to make it clearer for the reader.

P8 L3: observations

P8 L3: remove spacing and merge into one paragraph

P8 L13: deeper down along the profile, both variables

P8 L21: from the decomposition... likely, as shifts in ... decomposition are generally in the order of...

P8 L25: On the high saltmarsh... with depth also occurs.

P8 L26: ... characterised by a mixture of...

P8 L28: ...marsh growth Spartina anglica was also present at this ...

P9 L 22: remove spacing and merge into one paragraph

P9 L23: that determines

P10 L31: remove spacing, merge into one paragraph

Section 5: merge all into one paragraph.

―――――――――――――――――――――

---

## Referee Comment (RC2) · M. Schwartz (Referee) · 4 Oct 2016

The authors have presented a comprehensive assessment of both depositional and preservation factors influencing the accumulation of soil/sedimentary organic carbon across an estuarine salinity gradient. Their analysis of contributions from changes in surface vegetation type (e.g., C3 vs. C4 plants) and geochemical influence of OC decomposition rates at different salinity regimes provides a useful framework for assessing how forecast sea level rise could affect organic carbon storage in estuaries experiencing saltwater intrusion. Their examination of spatial variability in both OC supply and decomposition rates is robust and spans the estuarine salinity gradient.

Notable absent is data for (or an estimate of) sediment accretion rates at each of the

three estuarine zones sampled.

How will sea level rise and saltwater intrusion affect the location of the estuarine turbidity maximum and resulting allochthonous OC deposition?

---

## Author Comment (AC1) · 24 Oct 2016

**Replies on referee comments on *Van de Broek et al. (2016), The importance of an estuarine salinity gradient on soil organic carbon stocks of tidal marshes, Biogeosciences Discuss., doi: 10.5194/bg-2016-285***

**Referee #2 (M. Schwartz)**

*1. The authors have presented a comprehensive assessment of both depositional and preservation factors influencing the accumulation of soil/sedimentary organic carbon across an estuarine salinity gradient. Their analysis of contributions from changes in surface vegetation type (e.g., C3 vs. C4 plants) and geochemical influence of OC decomposition rates at different salinity regimes provides a useful framework for assessing how forecast sea level rise could affect organic carbon storage in estuaries experiencing saltwater intrusion. Their examination of spatial variability in both OC supply and decomposition rates is robust and spans the estuarine salinity gradient.*

We greatly thank dr. Schwartz for reviewing our manuscript and for his constructive comments.

*2. Notable absent is data for (or an estimate of) sediment accretion rates at each of the three estuarine zones sampled.*

This data is available in Temmerman et al. (2004, figure 8) and is added to section 2.1 (Study sites). For the period 1955 – 2002, the following average annual sediment accumulation rates are reported:

- Saltmarsh: about 0.75 and 0.5 cm yr$^{-1}$ for low and high marshes resp.
- Brackish marsh: about 1 – 2 and 0.5 – 1 cm yr$^{-1}$ for low and high marshes resp.
- Freshwater marsh: about 1 - 2 and 1 cm yr$^{-1}$ for low and high marshes resp.

*3. How will sea level rise and saltwater intrusion affect the location of the estuarine turbidity maximum and resulting allochthonous OC deposition?*

In section 4.4 (Discussion – Implications of sea level rise for estuarine soil organic carbon stocks) we state the maximum turbidity zone is predicted to shift more inland as a consequence of sea level rise. This will indeed effect which portion of the estuary receives a significant input of allochthonous (terrestrial) organic carbon, as this will also shift more upstream. We added to this section that as a consequence of the upstream migration of the maximum turbidity zone, terrestrial organic matter can travel less far downstream in the estuary. As a consequence, tidal marshes which are now located at the downstream end of the MTZ will receive less stable terrestrial OC in the future, which will decrease their potential to sequester OC, as in addition also the sedimentation rates will decrease as a result of the shifting location of the MTZ.

**References**

Temmerman, S., Govers, G., Wartel, S. and Meire, P.: Modelling estuarine variations in tidal marsh sedimentation: response to changing sea level and suspended sediment concentrations, Mar. Geol., 212(1–4), 1–19, doi:10.1016/j.margeo.2004.10.021, 2004.

---

## Author Comment (AC2) · 25 Oct 2016

**Replies on referee comments on *Van de Broek et al. (2016), The importance of an estuarine salinity gradient on soil organic carbon stocks of tidal marshes, Biogeosciences Discuss., doi: 10.5194/bg-2016-285***

**Anonymous Referee #1**

*1. General comments: The paper is well-written and generally well structured. It addresses an important gap in the field of carbon cycling, namely of measurements in brackish and freshwater marshes. The authors address various aspects clearly and draw attention to the problems associated with different sampling depths. In addition, they indicate what a future sea level rise may entail for the carbon storage dynamics within the Scheldt estuary.*

We thank the first anonymous referee for the comprehensive comments on our manuscript. These definitely improve the quality of our manuscript substantially. Below we provide answers to all comments.

*2. There are some aspects which need clarification and one main concern of mine is that samples were collected in different seasons. Depth profiles were collected in November whilst aboveground biomass was not collected until August. No mention of this is made in the discussion and I certainly believe that this needs to be addressed and justified.*

We agree with the reviewer that this point needs further clarification and should be discussed more extensively. Soil samples were collected at the beginning of this study, in November 2014. However, in order to calculate the total annual biomass produced at the locations where soil samples were collected the maximum annually produced biomass had to be collected. As it has been shown that the timing of peak standing biomass on tidal marsh in western Europe generally occurs in August (Groenendijk, 1984; De Leeuw et al., 1990), vegetation samples were not collected until August 2015. This is now clearly stated in the manuscript. This comment is further answered under question number 18.

*3. I also miss more discussion on the effect the very different types of vegetation may have on the carbon dynamics of the different marshes. This certainly influences stable isotope signatures and carbon mineralization rates. More comments are found in the specific comments regarding this.*

This issue is addressed under question number 18.

**Specific comments**

*4. P3 L3f: Why did the authors limit themselves to the incorporation of in situ produced belowground biomass? Aboveground biomass also produces substantial amounts of litter and can also be buried.*

Here we summarized the most important factors controlling the increase in elevation of tidal marsh platforms. The comments that aboveground biomass can also contribute to elevation changes after burial is certainly valid, and the sentence is adapted to also include aboveground biomass: '…and incorporation of in situ produced biomass (both above- and belowground) on the other hand'.

*5. P4 L18-20: the use of PSU/practical salinity unit is discouraged, nowadays salinity as written here would be unitless i.e. the authors should write: ": : :salt or polyhaline zone (salinity >18), brackish or mesohaline zone (salinity 5–18) and freshwater/oligohaline zone (salinity 0–5): : :"*

The PSU notation is removed from the manuscript.

*6. P5 L4: How were these samples stored during their transport i.e. were the 0.03 sections thus homogenized?*

The undisturbed soil cores were divided into 0.03m sections in the field. Every sample was stored in a minigrip bag and transported to the lab: soil samples from the different replicate cores (or from the same core) were never homogenized. This is added to the manuscript: 'The cores were divided into 0.03 m sections and every soil sample was stored in a reclosable bag in the field before transport to the lab.'.

*7. P5 section 2.2: why were depth profiles collected in November 2014 and aboveground biomass not until the end of August? How do the authors justify using data from such different seasons?!*

For the reply on this point we refer to the answers on comments 2 and 18.

*8. P5 Section 2.3: just make one paragraph for easier reading and change title to "Soil and biomass analysis"*

This is adjusted in the manuscript.

*9. P5 L18: What do the authors mean with split? This also needs clarification because now it sounds like only one of the five replicates was analysed. Is this the case, or are you describing what was done to each one of the five replicates? Please clarify.*

Indeed, only 1 out of 5 replicates of aboveground biomass was analyzed for C content and C isotopes. This is because every aboveground biomass sample was taken at a 0.5*0.5m area, thus representing the variability in biomass characteristics in an area were biomass is homogenous. The 'splitting' refers to the fact that all the biomass collected in this 0.5*0.5m surface was shredded and repeatedly divided into equal parts until only a small fraction was left. This subsample was analyzed in 3 replicates (and the average was calculated and reported). This is clarified in the manuscript: 'The total aboveground biomass of one of the replicates collected on a 0.25 m² surface area was shredded and repeatedly divided into equal parts until only a small portion was left. This was further grinded…'.

*10. P5 L25ff: The authors sectioned the cores into 0.03 m sections, so, when they say one sample every 0.09m, do they mean it is the sample at 0.06-0.09, or 0.09-0.12 and so forth? The same question applies to when they say every 0.18 m. Maybe rather say …For the other two replicate profiles every third sample was analysed (i.e. 0.06-0.09, 0.15-0.18, :…:, 0.69-0.72m) to a depth of 0.72m. Thereafter, a sample was analysed every 0.18 m."*

We thank the referee for pointing to this confusing formulation, this is clarified in the manuscript: 'At every location one soil profile was analysed in detail (every other depth sample, i.e. 0 - 0.03, 0.06 – 0.09m, … ).  For the other two replicate profiles every third sample was analysed (i.e. 0 – 0.03, 0.09 – 0.12, …) down to a depth of 0.72m. Thereafter, samples were analysed every 0.18 m.'.

*11. P5 L29: what linear interpolation technique was used to do this?*

In order to obtain organic carbon percentages for all depth intervals, the average organic carbon percentage at the depths at which three replicate samples were analysed (i.e. 0 – 0.03, 0.09-0.12m, …) was linearly interpolated. This way, we obtained continuous depth profiles of organic carbon percentage. The same method was used to obtain continuous depth profiles of bulk density. Both were necessary to calculate the total OC stocks at the study sites. This is clarified in the manuscript: 'Continuous depth profiles of OC percentage for layers of 0.01m were obtained using the average OC percentage at the depths at which three replicates were analysed (i.e. every 0.09 m). The OC percentages at these depths were linearly interpolated to obtain OC percentages for intermediate layers. Continuous depth profiles for bulk density were obtained in an identical way.'.

*12. P6 L6: Was only a check for normality done? Please also mention (and I hope the authors did!) that homogeneity of variance was also checked.*

Both a check for normality (Anderson-Daling test) and homogeneity of variance (Levene's test of equal variences) were performed. This is now mentioned in the manuscript in section 2.4 (Data Analysis).

*Please also specify what statistical were done since you mention differences in Figure 4? And please specify which level of probability was used (e.g. "with a level of significance of p<0.05)."*

The level of probability we used was 0.05, this is added to the manuscript (section 2.4). In section 2.4 we state that we checked for differences in mean biomass production rates using a one-way anova test after checking for normality with an Anderson-Darling test in Matlab. This is now complemented with the fact that we checked for homogeneity of variance using a Levene's test.

*13. P6 section 3.1: - also include here that detailed results for the grain size (not texture) are in the supplementary information*
This is included in the manuscript: '…with a silt loam grain size (detailed grain size data is provided in the Supplementary Information).'.

*14. Section 3.2:*
*- Figure S1 is not maximum annual biomass but as is noted in the figure caption as total biomass. This is a difference so please clarify.*
*- Even if the belowground data was not statistically analysed and no clear patterns are observed, I would have liked to see some comments on what we see i.e. that at the fresh low biomass is clearly very high, that for most sites we see very low values.*
*- An explanation is needed here for figure 4 and the letters apparently showing differences. These need to be explained.*

- 'Maximum' biomass is replaced by 'total' biomass in the manuscript
- The differences in belowground biomass production between the sites is now briefly discussed in section 3.2
- In section 3.2 (Results – Vegetation biomass production) we added the meaning of the different letters in figure 4, together with an interpretation of the different letters.

*15. P.7 section 3.4:*
*- Depth profiles of cumulative OC stock per 0.01 m layer are shown: … Where does this 0.01 m sectioning come from? The authors make no mention of this is in the methods. There you can only find 0.03 m sections or 0.1 and 0.2 m sections. Please clarify what I have missed.*
*- Please be more consistent with the terminology. Within this one paragraph the authors begin by using SOC but then use only OC later*

- The subdivision into layers of 0.01m depth was done using linear interpolation based on the depths with known OC%. This was done to graphically show how the total SOC stock up to a certain depth varies between the different sites. It is added to the material and methods section (2.4) that the linear interpolation was done for layers with a thickness of 0.01m. To section 3.4 it is added that the linear interpolation is described in section 2.4.
- The terminology of (S)OC was checked throughout the paper and adjusted where necessary. When it is not clear whether organic carbon in soils or in e.g. vegetation or deposited sediments is referred to, SOC was used. However, if from the context it is clear that we refer to organic carbon in soils, OC is was used.

*16. P7 L 11f: $\delta^{13}C$ signal of standing vegetation is closely related to the $\delta^{13}C$ signal of SOC in the topsoil layer. How is this conclusion reached? I presume with standing vegetation you mean the aboveground biomass? I would not agree with this from what I see in figure 6.*

We agree with the reviewer that the conclusion that the $\delta^{13}C$ signal of topsoil sediments is closely related to the $\delta^{13}C$ signal of the vegetation is drawn too easily and should be discussed in more

detail. Therefore, we adapted this sentence to provide a better overview of the observed relationships between topsoil and vegetation $\delta^{13}$C signals: '…SOC in the topsoil layer is similar to the $\delta^{13}$C signal of standing vegetation. However, close inspection shows that some differences in the $\delta^{13}$C signal between vegetation and topsoil can be observed. At the high freshwater marsh the topsoil $\delta^{13}$C signal is higher than the signal for both above- and belowground vegetation, while at the low freshwater marsh the topsoil $\delta^{13}$C signal is lower than the above- and belowground vegetation signal. At both the low and high brackish marshes, the topsoil $\delta^{13}$C is very similar to the $\delta^{13}$C signal of roots, while it is about 1‰ lower compared to the $\delta^{13}$C signal of aboveground vegetation. At the high saltmarsh, the topsoil $\delta^{13}$C signal has a value in between the $\delta^{13}$C signals of above- and belowground vegetation, while at the low saltmarsh the topsoil $\delta^{13}$C signal is significantly lower compared to the signal of both above- and belowground vegetation.'.

*17. P7 L18: "However, the differences reported in previous studies are almost always much smaller than the differences we find. This may to some extent be related to differences in environmental conditions, but differences in sampling procedures also matter." I agree that the authors want to address the problem of inconsistent sampling depth but I do not think that you can dismiss all the other reasons why there are such differences with this one sentence. The estuaries listed in Table 4 are all very different in terms of their geology, morphology, inputs, outputs, etc. and I would like to see some more discussion of this. One of the aims of this paper was to determine OC stocks along a salinity gradient of a temperate estuary and its main controls and I think this has to be addressed more thoroughly. Since the authors do actually discuss some of these factors in section 4.3, I would suggest that section 4.3 follows directly to 4.2 (or is combined) because the authors here try and further explain the observed patterns in SOC stocks which is a more natural progression from what is initiated in section 4.1. I would also bring the issue of different sampling depths then as a separate header and not as the first paragraph of the discussion. This is an aspect but not the most important one.*

*In relation to this it is unclear in line 20 whether the authors refer to differences from this study or from the other studies. This needs to be clarified.*

- We prefer not to merge sections 4.2 and 4.3, since in section 4.2 the observed patterns are discussed and interpreted, while in section 4.3 explanations for the observed patterns are discussed. We believe the manuscript would become less clear if the two sections would be merged.
- We agree with the reviewer that the issue of the effect of different sampling procedures should be discussed in a separate paragraph. To improve the structure of the manuscript we will start the discussion with the current section 4.2 (Observed patterns in SOC storage), followed by the section on the controlling factors (current section 4.3 – Explanations for the observed patterns in soil organic carbon stocks). The current section 4.1 (Soil organic carbon stocks along the estuary) will be discussed after this.
- The fact that differences in characteristics between the reported estuaries (e.g. environmental conditions and morphology) will have an effect on the reported SOC stocks in the cited studies is now briefly discusses in the manuscript. We will not discuss this in much detail, as this is not the goal of this study. However, based on the brief discussion the reader is aware of the fact that not only the effect of sampling procedure controls the reported OC stocks.
- In line 20 it is now indicated that these differences refer to the tidal marshes from other studies.

*18. Section 4.3.2: I miss a more thorough discussion on the fact that you have very different vegetation types. I presume no $\delta^{13}$C values are known for the different plants themselves?*

- We do have $\delta^{13}C$ for the different vegetation types (table S1). We agree with the reviewer that the discussion about the effect of vegetation types on the observed SOC stocks is limited. However, we do not have data to isolate the effect of different vegetation types on the observed SOC stocks along the estuary. Therefore, we complemented the discussion about the effect of vegetation (section 4.3.2) with observations that we made on the low and high portion of the same marsh. Based on this, the effect of vegetation on SOC stocks at the different marshes is now discussed in the manuscript:
    - Freshwater marsh: Although both the low and high marsh are characterized by different vegetation types (*P. australis* and *Salix* forest resp.) and annual biomass production is significantly different (much higher at the low marsh), depth profiles of OC% are remarkably similar. In addition, SOC stocks in the top 0.6m of the soil profile are higher on the high marsh. This shows that the impact of local vegetation on SOC stocks is limited at the freshwater marshes.
    - Brackish marsh: Both the low and high brackish marshes have the same vegetation type (*Elymus athericus*) and rates of annual biomass production are similar. However, both topsoil OC% (about 4% higher at the high marsh) and SOC stocks up to 0.6m depth (much higher for the high marsh) are significantly different. This again indicates that another factor besides local vegetation controls the size of the SOC stocks at these locations.
    - Saltmarsh: At the low marsh *Spartina anglica* is present. It has been shown before that *Spartina* vegetation is very labile and contributes little to the total SOC pool (Boschker et al., 1999; Bouillon and Boschker, 2006; Middelburg et al., 1997). At the high saltmarsh the C4 *Spartina* vegetation has been replaced by a community of C3 species. The OC concentrations in the top decimeters at the high saltmarsh is also higher compared to the low saltmarsh. This indicates that at these locations C3 vegetation species do contribute to the size of the SOC stocks.
- These observations indicate that local biomass production is probably not the dominant factor controlling SOC stocks along the estuary. It may however control local SOC stocks, as is the case on the saltmarshes.

*I also struggle with the fact that biomass was only measured in August, whilst all other measurements were taken in November. The influence of weather and climate conditions and subsequently river flow on affecting stable isotope signatures should not be underestimated (e.g. Zetsche et al. 2011, dx.doi.org/10.1016/j.csr.2011.02.006).*

We agree with the reviewer that the timing of soil sample collection can have an effect on the $\delta^{13}C$ signal of the top sediments, as shown by Zetsche et al. (2011). It should be noted that in Zetsche et al. (2011) only the top 0.01 m of sediments on a sandflat were analyzed, which are highly dynamic and characterized by both deposition and erosion. Zetsche et al. (2011) show that the intra-annual variations in the relative contribution of terrestrial-derived and marine C lead to changes in the $\delta^{13}C$ signal of the top 0.01 m sediments, which is not unexpected in such an environment.

Our study concerns tidal marsh sediments and here only deposition occurs. Our aim was to use the $\delta^{13}C$ signal of the whole soil profile, combined with the $\delta^{13}C$ signal from different inputs (allochthonous C and vegetation), to construct hypotheses on the origin of SOC in the studied tidal marshes. Therefore, the timing of soil sample collection will only be of minor importance, as the $\delta^{13}C$ signal at depth is an integration of the complete annual cycle of $\delta^{13}C$ variations over the past decades. We do agree that the $\delta^{13}C$ signal of the very top layer of the profiles we analysed may be affected by the same processes as those described by Zetsche et al. (2011). However, the variation of the contributions of terrestrial/marine/autochthonous C will not affect the deeper sediment layers. As our interpretations are based on the variation of the $\delta^{13}C$ signals over the whole profile we do not expect that this intra-annual variation to have a strong effect on our results and interpretations. We included this point in the discussion of the manuscript.

*I would suggest the authors also look at a recent similar study by Hansen et al. 2016 (DOI 10.1007/s11368-016-1500-8) and see how their results of the importance of salinity can be reconciled in this study also for section 4.3.1.*

We thank the reviewer for pointing to the recent study by Hansen et al. (2016). They also clearly show a decrease in tidal marsh SOC stocks with increasing salinity in another western European estuary (Elbe, Germany). We included the result from their study into our manuscript:

- We included their measurements of SOC stocks in tidal marshes in different salinity zones in Table 4.
- Hansen et al. (2016) was cited in the introduction among other studies reporting on estuarine SOC stocks and biomass production along an estuarine salinity gradient.
- We discussed the results of Hansen et al. (2016) in section 4.3.3, where we put forward arguments in order to explain the observed pattern in SOC stock along the estuary.

*19. P8 L7f: There is no relationship. Did you analyse this statistically? If so please provide test results here, or at least indicate (data not shown).*

Based on the fact that no relation was detected ($R^2$ = 0.004) it was chosen not to show the correlation. "($R^2$ = 0.004, Data not shown)" was included in the manuscript. If the reader wishes she/he can reconstruct the correlation analysis based on the data given in Table 3 and Table S2.

*20. P8 L19f: Elymus is considered an invasive species. Do you think it is invading here and will remain as the dominant vegetation type here? How will this affect influence SOC stocks in the future as conditions favour this plant?*

Van der Pluijm and De Jong (2008, in Dutch) indeed show that at least since 1980 this species occupies about 55-60% of the total marsh area at the studied brackish marsh, although the area it occupies did not increase significantly between 1980 and 2004. It is however not invasive on all marshes of the brackish portion of the estuary, as nearby marshes are occupied dominantly with e.g. Phragmites australis. Therefore it is difficult to predict whether or not this species will invade other marshes in the future, e.g. due to changing environmental conditions or sea level rise.

Based on our data we cannot assess how this species will influence SOC stocks after invasion, as we only have data for brackish marshes under Elymus vegetation. We have knowledge of only 1 study that assessed the effect of establishment of *Elymus athericus* on SOC stocks by Valery et al. (2004), who show that over a period of 10 years after establishment of *Elymus*, no significant changes in sediment C concentrations were found. However, as they showed that *Elymus* litter contained significantly more lignin compared to the former vegetation, the tidal marsh changed from a source to a sink of C due to the low mineralization rates of *Elymus* litter. Based on the results from Valery et al. (2004), we do not expect changes in the SOC stock after *Elymus* establishment on a short timescale (10 years), however, increasing SOC stocks can be expected as relatively resistant *Elymus* litter will be incorporated in the marsh sediments.

**Figures**

*21. Personally I would prefer it if the authors used the blue colours always for the saltmarshes (since closest to the blue ocean) and the green colour for the freshwater marshes (closest to land) in the figures. This is more intuitive to the reader.*

This is a good suggestion which will increase the readability of the figure, this is adjusted in the manuscript.

*22. Figure 1: Please increase the font size of the country names in the inset. FYI: A black and white version of the map will not depict the light grey areas.*

The font size of the country names is increased (and repositioned, as they appear to have shifted).

*23. Figure 2: Brackish water marsh not just Brackish marsh*

We prefer to keep the term 'brackish marsh' throughout the manuscript and also in the figures. This term is also used in other studies (e.g. Hansen et al. (2016), Callaway et al. (2012), Dausse et al. (2012))

*24. Figure 3: All species names should be italicized. Figure caption: At several marshes the former tidal sandflat was reached, whilst at two other locations the marsh sediments extended below the maximum sampling depth of 1.4 m. The vegetation history is based on Temmerman et al. (2003) and information from the $\delta^{13}C$ profiles of this study, in combination with information from Boschker et al. (1999) and Middelburg et al. (1997). Mix denotes a mixed vegetation which included the following species…. A '?' indicates that no clear identification was possible.*

The species names are italicized. We added the species types that 'Mix' denotes. We also included that a '?' after a vegetation species denotes that the presence of this species is hypothesized while a '?' at a dashed line denotes that there is uncertainty concerning the exact depth of the vegetation transition.

*It is not possible to say only shallow marshes because the sandflat is also reached at the high saltmarsh and I presume only freshwater and brackish water high went beyond 1.4 m? Also specify what mix stands for. The figure has to be understandable on its own.*

We changed the sentence 'At shallow marshes the former tidal sandflat was reached, at other locations the marsh sediments extended below the maximum sampling depth of 1.4 m.' into: 'At locations where the sandflat was reached this is indicated, at the other locations the marsh sediments extended below 1.4m depth.' Also, a sandflat layer will be added below the low freshwater marsh.

*25. Figure 4: the inset is very distracting. Please remove. Instead you can insert a break on the y-scale to allow the details to be seen more easily for the belowground biomass. Adjust the figure caption i.e. remove "(the inset… .biomass)". Also add the y-axis legend i.e. Biomass production (g dry weight m$^{-2}$ yr$^{-1}$). Replicas should be replicates. The letters to indicate significant differences are confusing. It has to be explained in the figure caption what the different letters stand for. No mention of these are made in the main text which also has to be addressed!*

The inset is removed, an y-axis break is inserted and the caption is changed accordingly. The reason why this wasn't done is that this way the figure increases in height. Furthermore, we will add a y-label. We also explained the different letters in the caption and in the main text.

*26. Figure 5: Error bars for specific depths represent the standard deviation.*

This is changed in the caption.

*27. Figure 6: aboveground (circles)… Error bars represent the standard deviation.*

These changes are made in the caption

*28. Figure 7: write out OC once as organic carbon in the figure caption.*

This is adapted

**Tables**

*29. Comments like A, B, C etc. should be added as footnotes. They are footnotes and should not be in the main caption text.*

This is changed in all figures containing comments (A, B, …)

*30. Table 1: please change around C and D (better to have A, B, C in the same line and then D at the bottom for the mixed vegetation. Please also italicize all species names in the footnote D (previously footnote C). Regarding footnote C (previously D): What is texture? It is not texture but grain size that was measured in this study. Why is this called maximum marsh sediment depth? I would rather simply write "Maximum sampling depth". The tidal sandflat that is reached most likely is deeper but probably caused problems with the sampling device? Sand is not easy to sample.*

The letters C and D are changed, and species names italicized. In the caption, 'texture' will be changed to 'grain size'.

We named this 'maximum marsh sediment depth' because at this depth there was a transition from the silt/clay marsh sediments to the sandy former mudflat sediments. At the locations where we cored down to the sandy layer we were always able to collect at least the upper 10cm of sand (deeper sandy sediments were indeed difficult to sample). These sandy layers were also analyzed for grain size and OC content, which also allowed us to delineate the marsh/mudflat boundary based on these depth profiles. We prefer to keep the term 'maximum marsh sediment depth' because this informs the reader on the thickness of the marsh sediments at the sample locations. We explained this better in section 3.1 (Results – Soil characteristics).

*31. Table 2: Keep footnotes C and D and make them A and B. Add to figure caption: "Bulk density values are averages for the upper meter of soil, whilst soil pH and electrical conductivity were measured in the topsoil only.*

This is changed in table 2.

*32. Table 3: Increase the space between the line termed saltwater and the next line for 'up to 0.6 m depth' to make this clearer for the reader. Figure caption: Total organic carbon (OC) stock (kg… deviations calculated for the full vertical sampling profiles (depths used for the calculations are given in brackets), and the upper 0.6 m.*

The line spacing is increased, and the caption changed according to the comments

*33. Table 4: make this into a horizontal table and thus more readable. Perhaps place the location then as a separate column next to the estuary name.*

We changed the table to a horizontal layout, and will add an additional column for the location of the estuary if this does not makes table too wide. In addition, the OC stocks as measured by Hansen et al. (2016) were added to this table.

**Supplemental data**

*34. I would welcome that the excel sheets provided in the supplemental data are at least referred to in the paper.*

The excel tables are now referred to in the paper: the texture data is be referred to in section 3.1 (Resuls – Soil characteristics), the OC, CN and $\delta^{13}$C data in section 3.3 (Result – Soil organic carbon depth profiles).

*35. Figure S1: see my comments on Figure 4. Please also remove the inset here.*

Figure S1 is adapted in the same was as Figure 4, i.e. the inset is removed, an y-break will be placed and the caption adapted.

*36. Figure S2: why is there now mention of a depth interval of 0.01m? This is never mentioned previously in this study, only slicing at 0.03 m and 0.1 +0.2 m intervals is ever mentioned. Please explain.*

The 0.01 m depth intervals are based on interpolation, we refer to our answer on comment 15, where we explain why and how this was done.

*37. Table S3: Please italicize all species names. Replace Oosterschelde with Eastern Scheldt and Westerschelde with Western Scheldt.*

The species names are italicized, and Oosterschelde and Westerschelde put in English.

*38. Table S2: Figure caption: Average values (±SD) for aboveground, belowground (maximum root depth is given in brackets (m)) and total biomass, biomass production, organic carbon and nitrogen concentration (%), C:N ratio as well as the δ¹³C signal (‰ for vegetation at the study sites. Remove footnote A, footnote B: write here in full as a footnote the species. In table: Adjust either DW or dry weight, now have both. Also write species names in full. If you miss space you can shorten Freshwater to Fresh, etc. and add to caption "…at the study sites (freshwater, brackish water and saltwater marshes)."*

The figure caption is adjusted based on the suggestion of the reviewer. We addes a footnote B where the species at the high saltmarsh are listed. We consistently changed 'dry weight' into 'DW' in the table, and species names are now written in full.

**Technical corrections**

We greatly thank the reviewer for the detailed technical comments that will contribute greatly to the quality of the manuscript. The comments that are listed below without an answer are changed in the manuscript. Answers to technical comments that require explanation are given below as well.

*P2 L14: downstream of the maximum…*
*P3 L2: replace extratropical with temperate. Extratropical is not normally used in this context.*

We chose the term 'extratropical', since tidal marshes also occur in other climate zones. Therefore, we propose to change this sentence to 'These are vegetated intertidal areas located along coastlines and estuaries of sub-Arctic to tropical climates, although they occur mostly in temperate zones, and are among the most productive ecosystems on Earth'.

*P3 L7: equilibrium with the local*
*P3 L8: remove 'in particular'*
*P3 L16-17: remove spacing and merge into one paragraph.*
*P3 L22: tidal marshes, for which no data is available, is the*
*P3 L23-24: remove separation into paragraphs. These three reasons are all one aspect and should be together in one paragraph.*
*P3 L25: …(Craft, 2007). A sharp increase in salinity…*
*P3 L29: …2010). In addition, the OC input in tidal marsh…*
*P3 L32: data not date*
*P4 L5: remove space and form one paragraph.*
*P4 L8: …stocks in tidal marsh soils. The aims…*
*P5 L6f: …0.5m depth, and then in 0.2 increments down to the maximum depth of 1.4m.*
*P5 L17 and L23: replace weighted with weighed. Samples were placed on a scale, hence they were weighed. Weighted is used in a different context.*
*P5 L19: …using the Elemental Analyser…*
*P5 section 2.4: remove line spacing and form one paragraph.*
*P5 L26: …analysed to a depth of 0.72m. Below this depth, samples were analysed every 0.18 m.*
*P6 L5: remove "is"*

*P6 L17: willow trees were*
*P6 L18: what is meant by woody parts, this is not a correct term!*

With 'woody parts' we meant the standing vegetation of willow trees. We changed this sentence into: '…, while standing willow vegetation could not be collected…'.

*P6 L19: deduced from other studies*
*P6 L23: showed and decreased i.e. past tense.*

We prefer to keep the results section in the present tense.

*P6 L26: do not write just in the top of the profile, be more specific, e.g. " …OC concentration in the upper 0.2 m." Or whichever depth it is…*
*P7 L2: to the low marshes*
*P7 L11: this is the first time a 'C4 Spartina site' is mentioned, please refer to this differently to make it clearer for the reader.*

This sentence is changed to: 'For all sites except the low saltmarsh, which is characterised with *Spartina anglica* vegetation (C4 type), the $\delta^{13}$C signal…'.

*P8 L3: observations*
*P8 L3: remove spacing and merge into one paragraph*
*P8 L13: deeper down along the profile, both variables*
*P8 L21: from the decomposition: : : likely, as shifts in … decomposition are generally in the order of…*
*P8 L25: On the high saltmarsh: : : with depth also occurs.*
*P8 L26: … characterised by a mixture of…*
*P8 L28: …marsh growth Spartina anglica was also present at this …*
*P9 L 22: remove spacing and merge into one paragraph*
*P9 L23: that determines*
*P10 L31: remove spacing, merge into one paragraph*
*Section 5: merge all into one paragraph.*

**References**

Boschker, H. T. S., de Brouwer, J. F. C. and Cappenberg, T. E.: The contribution of macrophyte-derived organic matter to microbial biomass in salt-marsh sediments: Stable carbon isotope analysis of microbial biomarkers, Limnol. Oceanogr., 44(2), 309–319, doi:10.4319/lo.1999.44.2.0309, 1999.

Bouillon, S. and Boschker, H. T. S.: Bacterial carbon sources in coastal sediments: a cross-system analysis based on stable isotope data of biomarkers, Biogeosciences, 3, 175–185, doi:10.5194/bg-3-175-2006, 2006.

Callaway, J. C., Borgnis, E. L., Turner, R. E. and Milan, C. S.: Carbon Sequestration and Sediment Accretion in San Francisco Bay Tidal Wetlands, Estuaries and Coasts, 35, 1163–1181, doi:10.1007/s12237-012-9508-9, 2012.

Dausse, A., Garbutt, A., Norman, L., Papadimitriou, S., Jones, L. M., Robins, P. E. and Thomas, D. N.: Biogeochemical functioning of grazed estuarine tidal marshes along a salinity gradient, Estuar. Coast. Shelf Sci., 100, 83–92, doi:10.1016/j.ecss.2011.12.037, 2012.

Groenendijk, A. M.: Primary production of 4 dominant salt-marsh angiosperms in the southwestern Netherlands, Vegetatio, 57(2/3), 143–152, 1984.

Hansen, K., Butzeck, C., Eschenbach, A., Gröngröft, A., Jensen, K. and Pfeiffer, E. M.: Factors influencing the organic carbon pools in tidal marsh soils of the Elbe estuary (Germany), J. Soils Sediments, 1–14, doi:10.1007/s11368-016-1500-8, 2016.

De Leeuw, J., Olff, H. and Bakker, J. P.: Year-to-Year variation in peak above-ground biomass of six salt-marsh angiosperm communities as related to rainfall deficit and inundation frequency, , 36, 139–151, 1990.

Middelburg, J. J., Nieuwenhuize, J., Lubberts, R. K., van de Plassche, O. and Vandeplassche, O.: Organic carbon isotope systematics of coastal marshes, Estuar. Coast. Shelf Sci., 45, 681–687, doi:10.1006/ecss.1997.0247, 1997.

Van der Pluijm, A. M. and De Jong, D. J.: Vegetatieontwikkeling westelijk deel Schor van Waarde (Westerschelde) 1981 - 2006, Middelburg., 2008.

Valery, L., Bouchard, V. and Lefeuvre, J. C.: Impact of the invasive native species Elymus athericus on carbon pools in a salt marsh, Wetlands, 24(2), 268–276, doi:10.1672/0277-5212(2004)024[0268:IOTINS]2.0.CO;2, 2004.

Zetsche, E., Thornton, B., Midwood, A. J. and Witte, U.: Utilisation of different carbon sources in a shallow estuary identified through stable isotope techniques, Cont. Shelf Res., 31(7–8), 832–840, doi:10.1016/j.csr.2011.02.006, 2011.